

# The University of Victoria Cloud Feedback Emulator (UVic-CFE): cloud radiative feedbacks in an intermediate complexity model

Ullman, D.J.[1,*] and Schmittner, A.[1]

[1]College of Earth, Ocean, and Atmospheric Sciences, Oregon State University, USA
[*]Now at Northland College, Ashland, WI, USA

*Correspondence to*: David J. Ullman (dullman@northland.edu)

**Abstract.** The dominant source of inter-model differences in comprehensive global climate models (GCMs) are cloud radiative effects on Earth's energy budget. Intermediate complexity models, while able to run more efficiently, often lack cloud feedbacks. Here, we describe and evaluate a method for applying GCM-derived shortwave and longwave cloud feedbacks
from $4xCO_2$ and Last Glacial Maximum experiments to the University of Victoria Earth System Climate Model. The method generally captures the spread in top-of-the-atmosphere radiative feedbacks between the original GCMs, which impacts the magnitude and spatial distribution of surface temperature changes and climate sensitivity. These results suggest that the method is suitable to incorporate multi-model cloud feedback uncertainties in ensemble simulations with a single intermediate complexity model.

**1 Introduction**

The predominant trade-off in climate modeling is that of systematic complexity versus computational expense. While comprehensive global climate models (GCMs) attempt to resolve the complex interactions between earth systems, their computational expense limits the exploration of parametric uncertainty. Conversely, more simplified models, such as earth system models of intermediate complexity (EMICs), can be employed for large-ensemble analysis of parametric variability
but their reliance on fixed boundary conditions or generalized parameterizations of earth processes may not capture all important feedbacks driving system dynamics.

One of the largest sources of intermodel spread in GCM-based climate projections is the magnitude and direction of radiative cloud feedbacks (Soden and Held, 2006; Dufrense and Bony, 2008; Tomassini et al., 2013; Vial et al., 2013). Clouds affect climate through their impacts on both shortwave (solar radiation mostly in the visible part of the spectrum) and longwave
(terrestrial, infrared radiation) fluxes and therefore determine the sensitivity of GCMs to changes in radiative forcing (Andrews et al., 2012; Sherwood et al., 2014). Because clouds are more reflective than most other surfaces, an increase in clouds will reduce the amount of shortwave energy absorbed by the Earth and lead to cooling. Conversely, clouds ability to absorb upward longwave fluxes and re-radiate them back down causes warming at the surface (Hartmann and Short, 1980). The relative magnitude and net effect of these feedbacks depends on cloud altitude. For low clouds, which radiate longwave fluxes at a



similar temperature as the surface, shortwave effects dominate and their net effect is cooling. High clouds, on the other hand, radiate at much colder temperatures than the surface, which can make the longwave effect dominate and lead to net warming (Hartmann et al., 1992). The spread in GCM cloud feedbacks is primarily driven by model differences in low cloud cover changes (Sherwood et al., 2014). In addition, the spread in GCM cloud feedbacks manifests in both the global mean as well as

regional variability (Tomassini et al., 2013; Vial et al., 2013). This spatial variability likely has a profound impact on the magnitude of climate response to perturbations (Marvel et al., 2015).

Since EMICs use simplified atmospheric components, the cloud radiative forcing is typically fixed (Plattner et al., 2001; Joos et al., 2001; Driesschaert, 2005; Crucifix et al., 2002; Weaver et al., 2001). Therefore, the uncertainties in cloud feedbacks demonstrated in GCMs are typically neglected in the non-interactive cloud schemes of EMICs. Schmittner et al.

(2011), e.g., hypothesized that their estimate of climate sensitivity, determined using the University of Victoria (UVic) EMIC and paleoclimate observations, resulted in a too narrow probability distribution due to the neglect of cloud feedback uncertainties. Here we describe and evaluate a new method for diagnosing and applying cloud feedbacks of state-of-the-art GCMs into an EMIC, thereby creating a computationally less-expensive emulator of more complex models.

## 2 Methods

### 2.1 Model Description

The UVic Earth System Climate Model (Weaver et al., 2001) is an EMIC with a 3-dimensional ocean general circulation model coupled to a dynamic-thermodynamic sea ice model, a 2-dimensional single-layer energy-moisture balance atmosphere, and a dynamic land (Meissner et al., 2003) and vegetation model (Cox, 2001). Surface wind speeds used in the calculations of air-sea exchange and atmospheric transport of heat and moisture are prescribed in the model, thereby limiting variability in the

atmospheric model. The model conserves heat and moisture without the need for a flux correction (Weaver et al., 2001). We employ version 2.9 of UVic (Eby et al., 2013), in which atmospheric heat diffusion varies with changes in global mean surface air temperature; this modification has been shown to improve the latitudinal temperature gradient for the Last Glacial Maximum when compared with high-latitude proxy data (Fyke and Eby, 2012). All model components have a horizontal grid resolution of 1.8° latitude by 3.6° longitude, with 19 vertical levels in the ocean model increasing from 50 m thickness in the

surface level to 590 m thickness in the deepest gridcell.

The net radiative balance (NETRAD) at the top of the atmosphere (TOA) is the difference between the net shortwave radiation ($SW_{TOA}$) and the outgoing longwave radiation ($LW_{TOA}$):

$$NETRAD = SW_{TOA} - LW_{TOA} \tag{1}$$


Clouds impact $SW_{TOA}$ through prescribed monthly fields of atmospheric albedo ($\alpha_{atm}$):





$$SW_{TOA} = S - S \cdot \alpha_{atm} - S \cdot (1 - \alpha_{atm}) \cdot \alpha_{sfc} \cdot \tau^2 \qquad (2)$$

where S is the flux of incoming (incident) solar radiation energy per unit area (W m$^{-2}$) at the top of the atmosphere (with seasonal and latitudinal variation), $\tau$ is a constant atmospheric transmission coefficient (0.77), and $\alpha_{sfc}$ is the surface albedo. The second term of Eq. (2) represents the proportion of incoming SW radiation that is immediately reflected by clouds, while the third term represents the portion that is reflected by the surface, which passes through the atmosphere twice. During its first (downward) pass $S \cdot (1 - \alpha_{atm}) \cdot (1 - \tau)$ is absorbed by the atmosphere. During its second (upward) pass, $S \cdot (1 - \alpha_{atm}) \cdot (1 - \tau) \cdot \alpha_{sfc} \cdot (1 - \tau)$ is absorbed. All variables except for $\tau$ vary over space and time, but while $\alpha_{sfc}$ is allowed to evolve with changes in surface model components (sea ice, snow cover, vegetation, etc.), $\alpha_{atm}$ is a fixed boundary condition at monthly resolution to resolve seasonal changes in regional cloud cover. In the control version of UVic, $\alpha_{atm}$ is estimated with the following relationship:

$$\alpha_{atm} = \frac{f \cdot \alpha_{plt} - \alpha_{sfc}}{1 - \alpha_{sfc} \cdot \tau^2} \qquad (3)$$

where

$$\alpha_{plt} = \frac{SW_{out,TOA}}{SW_{in,TOA}} \qquad (4)$$

$$\alpha_{sfc} = \frac{SW_{out,sfc}}{SW_{in,sfc}} \qquad (5)$$

where the planetary ($\alpha_{plt}$) and surface albedo ($\alpha_{sfc}$) are calculated using the incoming and outgoing shortwave satellite observational measurements at the surface and top of the atmosphere from the Earth Radiation Budget Experiment (ERBE; Barkstrom, 1984; Barkstrom and Smith, 1986; Ramanathan et al., 1989). This $\alpha_{atm}$ relationship is directly derived from Eq. (2) so as to be internally consistent with the radiative balance relationship from the UVic model. The variable $f$ in Eq. (3) is a constant planetary albedo adjustment factor to account for radiative imbalances that arise in the implementation of the derived $\alpha_{atm}$.

The outgoing longwave radiation ($OLW$) is parameterized in UVic using an empirical relationship (Thompson and Warren, 1982; Weaver et al., 2001) that depends on surface relative humidity ($rh$) and temperature ($SAT$):

$$OLW = c_{00} + c_{01}rh + c_{02}rh^2 + (c_{10} + c_{11}rh + c_{12}rh^2)SAT$$
$$+ (c_{20} + c_{21}rh + c_{22}rh^2)SAT^2 + (c_{30} + c_{31}rh + c_{32}rh^2)SAT^3$$



$$+\Delta F_{2xCO_2} ln \frac{[CO_2]_t}{[CO_2]_o} \tag{6}$$

where the final term adjusts *OLW* for a change in the atmospheric $CO_2$ concentrations. The value of $\Delta F_{2xCO_2} = 5.35$ W m$^{-2}$ is selected as the radiative forcing associated with 3.71 W m$^{-2}$ (IPCC, 2001). The constants ($c_{xx}$) are provided by Thompson and Warren (1982). Since this is an empirical relationship, the effect of clouds on the LW radiative balance is not explicitly but implicitly included.

## 2.2 CERES update to Atmospheric Albedo boundary conditions

Because of discontinuities in satellite coverage, missing data, and poor resolution, the Clouds and the Earth's Radiant Energy System (CERES, Wielicki et al., 1996) was launched in late 1999 to better observe the earth's radiative balance (Fasullo and Trenberth, 2008). The CERES experiment uses an updated satellite architecture and provides higher spatial resolution observations over a longer time domain (2000-2013 for CERES compared with 1985-1989 for ERBE), thereby providing more robust modern climatology on the impact of clouds on atmospheric albedo (Wielicki et al., 1996). In addition, the duration of the ERBE experiment between 1985 to 1989 spans a somewhat large El Niño event (1987), which may bias the equatorial Pacific toward enhanced cloudiness in the calculation of atmospheric albedo climatology using the ERBE data (Cess et al., 2001).

In this paper, we use the climatology (2000-2013) of CERES surface and top of the atmosphere shortwave fluxes to better estimate $\alpha_{atm}$ boundary conditions in UVic (using Eq. (3)). A value of $f = 0.95$ in Eq. (3) was selected in order to match 20$^{th}$ century global mean temperature data estimates of ~13.9 °C (NOAA, 2016) in a UVic control simulation. The final estimate of CERES-based $\alpha_{atm}$ is smoothed and regridded to the UVic grid.

Figure 1 compares the annual mean values of $\alpha_{atm}$ as derived from the ERBE and CERES datasets. In the tropics, the ERBE-based estimates of $\alpha_{atm}$ generally match those of the CERES-based values (Figure 1). In the high latitudes, however, the ERBE-based $\alpha_{atm}$ values are generally higher than the CERES-based values. Such differences are likely related to improvement in sampling orbit of the CERES satellite and the associated reduction in zenith angle-dependent biases, which may result in large errors in the top of the atmosphere flux measurements in the ERBE data (Loeb et al., 2009). As such, the use of CERES-based estimates of $\alpha_{atm}$ provides an improvement in UVic, particularly at high latitudes.

## 2.3 Innovations

With the use of CERES-based $\alpha_{atm}$ estimates, the UVic model now includes an updated effect of clouds on the Earth's shortwave radiative balance. However, the control UVic model design does not incorporate any change in the shortwave or longwave radiative effect of clouds due to changes in temperature. This lack of cloud feedbacks may significantly limit the ability of UVic to capture global temperature in perturbed simulations. Here, we provide a simple method of diagnosing cloud radiative forcings from GCM results of the Coupled Model Intercomparison Project 5 (CMIP5) and Paleoclimate Model





Intercomparison Project 3 (PMIP3) archives (Braconnot et al., 2011; Taylor et al., 2012) and incorporate the associated shortwave and longwave cloud feedbacks into UVic for both 4 times $CO_2$ (4x$CO_2$) and Last Glacial Maximum (LGM) climate simulations. Reanalysis of satellite observations suggests that the range of CMIP5 models present widespread agreement with cloud data, both in spatial extent and vertical distribution, across the historical record (Norris et al., 2016). We have selected

model output from 7 GCMs: CCSM4 (abbreviated as CCSM), CNRM-CM5 (CNRM), GISS-E2-R (GISS), IPSL-CM5A-LR (IPSL), MIROC-ESM (MIROC), MPI-ESM-P (MPI), and MRI-CGCM3 (MRI). These models were chosen because they have results for both 4x$CO_2$ and LGM simulations and all of the relevant variables for calculating shortwave and longwave cloud feedbacks (see below). The following innovations demonstrate how we employ UVic as a cloud feedback emulator (UVic-CFE) of the full AOGCMs.

**2.3.1 Shortwave Cloud Feedbacks in UVic**

Since UVic incorporates the shortwave impact of clouds through atmospheric albedo, we assess the shortwave cloud feedback as the change in $\alpha_{atm}$ due to the change in temperature in each of the GCM simulations. Albedo anomalies are not mathematically additive; therefore, we first calculate $\alpha_{atm}$ for each perturbed state (4xCO2, LGM) by adding GCM anomalies of each of the individual fluxes to the CERES observations:

$$SW_{in,TOA,GCM} = (SW_{in,TOA,perturbed} - SW_{in,TOA,control}) + SW_{in,TOA,CERES} \tag{7}$$

$$SW_{out,TOA,GCM} = (SW_{out,TOA,perturbed} - SW_{out,TOA,control}) + SW_{out,TOA,CERES} \tag{8}$$

$$SW_{in,sfc,GCM} = (SW_{in,sfc,perturbed} - SW_{in,sfc,control}) + SW_{in,sfc,CERES} \tag{9}$$

$$SW_{out,sfc,GCM} = (SW_{out,sfc,perturbed} - SW_{out,sfc,control}) + SW_{out,sfc,CERES} \tag{10}$$

For each of the variables, we have calculated a 12-month climatology (separate averaging for each month) that is assessed over

the final 10 years of the 150 year transient 4x$CO_2$ simulations, the final 100 years of the LGM equilibrium simulations, and the final 100 years of the equilibrium control simulations. The anomaly-perturbed values of each of the shortwave fluxes (Eq. (7)-(10)) are then used to calculate an $\alpha_{atm,perturbed}$ for each of the perturbed GCM simulations using Eq. (3)-(5).

Again, because albedo values are not additive, we calculate the albedo anomaly as the ratio of the atmospheric albedo of the GCM perturbed state to CERES-derived atmospheric albedo. Therefore, the $\alpha_{atm}$ feedback ($\alpha_{atm}FB$) is this albedo

anomaly divided by the change in temperature:



$$\alpha_{atm}FB = \frac{\left[\alpha_{atm,perturbed}/\alpha_{atm,CERES}\right] - 1}{SAT_{perturbed} - SAT_{control}} \qquad (11)$$

The subtraction of 1 in the numerator is necessary such that when there is no change in $\alpha_{atm}$ ($\alpha_{atm,perturbed} = \alpha_{atm,CERES}$), then there is no atmospheric albedo feedback. This $\alpha_{atm}$ feedback is calculated as a 12-month climatology for each of the 7 GCMs

that are sampled in this analysis (Figure 2, 3). Positive (negative) values for this atmospheric albedo feedback indicate a negative (positive) shortwave cloud feedback since increases in temperature cause an increase (decrease) in atmospheric albedo, which cools (warms) the surface. The magnitude of these atmospheric albedo feedbacks varies considerably among the GCMs and between perturbed climate states ($4xCO_2$ versus LGM), which is consistent with the large spread in cloud shortwave feedbacks found in previous studies (Tomassini et al., 2013; Vial et al., 2013). For example, GISS-E2-R shows a

strongly positive atmospheric albedo feedback from the $4xCO_2$ results, while IPSL-CM5A-LR generally shows a strongly negative atmospheric albedo feedback, particularly in the tropics (Figure 2).

The innovation to UVic is the application of these GCM-diagnosed $\alpha_{atm}$ feedbacks to the shortwave radiative balance. First, we calculate a SAT climatology from a long-term control simulation of UVic that uses $\alpha_{atm,CERES}$ as the control atmospheric albedo. Then at each timestep (t) of a model simulation, we calculate the difference in surface air temperature

from this control climatology, and perturb atmosphere albedo using the GCM-derived $\alpha_{atm}FB$ of Eq. (11):

$$\alpha_{atm}(t) = \left[\alpha_{atm}FB \cdot [SAT(t) - SAT_{ctl}] + 1\right] \cdot \alpha_{atm,CERES} \qquad (12)$$

The above calculation is done at every timestep and each gridcell allowing for spatially- and seasonally-specific atmospheric

albedo feedbacks as diagnosed from the GCMs.

### 2.3.2 Shortwave Cloud Feedbacks in UVic

Because UVic lacks a longwave cloud feedback in the calculation of *OLW*, we provide an additional term to Eq. (6), which now includes the *OLW* due to changes in the cloud longwave effect in the GCM simulations. First, we diagnose the outgoing longwave radiation at the top of the atmosphere from the GCM output:


$$OLW_{cloud} = OLW_{total} - OLW_{clearsky} \qquad (13)$$

The outgoing longwave cloud feedback is therefore the cloud longwave forcing anomaly divided by temperature anomaly:

$$OLW_{cloud}FB = \frac{OLW_{cloud,perturbed} - OLW_{cloud,control}}{SAT_{perturbed} - SAT_{control}} \qquad (14)$$



as diagnosed from results of the GCM perturbed simulations. These outgoing longwave cloud feedbacks are calculated separately for both the 4xCO$_2$ and LGM perturbed states (Figure 4, 5). Most models show more areas of positive $OLW_{cloud}FB$. This indicates a negative climate feedback since increasing temperatures lead to more $OLW$, which cools the surface. Again, the outgoing longwave cloud feedbacks vary considerable between models and climate state. The largest variability in OLW

cloud feedbacks between models exists in the tropics, which is consistent with prior results suggesting that model differences in convective mixing and resulting cloud height greatly impacts the magnitude and direction of cloud feedbacks (Sherwood et al., 2014). Generally, the OLW cloud feedback is stronger in magnitude for the LGM state (Figure 5) than for the 4xCO$_2$ state.

Similar to the inclusion of the atmospheric albedo feedbacks in UVic, we multiply the outgoing longwave cloud feedback by the temperature difference from the long-term control UVic simulation:

$$OLW_{cloud}(t) = OLW_{cloud}FB \cdot [SAT(t) - SAT_{ctl}] \qquad (15)$$

This $OLW_{cloud}$ term is calculated at each timestep in the model and is added to the $OLW$ parameterization (Eq. (6)) as an additional cloud longwave feedback term.

**2.4 Numerical Experiments**

To estimate how well our UVic cloud feedback emulator (UVic-CFE) captures the original cloud radiative effects from the GCMs, we present an ensemble of UVic-CFE control and perturbed experiments (4xCO$_2$ and LGM) that use the $\alpha_{atm}$ and $OLW_{cloud}$ feedbacks diagnosed from each of the 7 GCMs employed in this analysis. Because our diagnosed cloud feedbacks differ between the 4xCO$_2$ and LGM climate states (Figures 2-5), we ran 2 separate preindustrial control simulations for each

ensemble member: one with 4xCO$_2$ cloud feedbacks (ctl4x) and one with LGM cloud feedbacks (ctlLGM). Indeed, the inclusion of these cloud feedbacks in the control climate state leads to slight differences in control global mean temperature, indicating that separate controls are necessary in the calculation of resulting radiative feedbacks. Therefore, we present the results from 28 separate UVic-CFE simulations: 4 simulations (ctl4x, ctlLGM, 4xCO$_2$, LGM) for each of the 7 GCM-derived cloud feedbacks.

Preindustrial control and LGM simulations with each of the GCM-derived cloud feedbacks were run to extended equilibrium (>2000 years) to be certain of minimal model drift (global mean SAT trend < 0.04 °C per 100 years). Both 4xCO$_2$ and LGM simulations follow the CMIP5/PMIP3 protocol (Braconnot et al., 2011; Taylor et al., 2012) as closely as possible as these are the boundary conditions used in the original GCM simulations. Our 4xCO$_2$ simulations using modern boundary conditions, an instantaneous increase in atmospheric CO$_2$ concentration to 1120 ppm, and a simulation length of 150 years,

starting from the end of the preindustrial control simulation (ctl4x). Our LGM simulations have reduced greenhouse gas concentrations (atmospheric CO$_2$ = 185 ppm; radiative forcing adjusted for appropriate CH$_4$/N$_2$O concentrations; Schmittner et al., 2011), altered orbital state, full glacial ice sheet extent/topography (Peltier, 2004), modified river pathways, and +1 PSU



(Practical Salinity Unit) increase in mean ocean salinity. In addition, we apply LGM surface wind stress anomalies that are diagnosed from the LGM GCM results (Muglia and Schmittner, 2015).

## 3 Results

### 3.1 Assessment of GCM-diagnosed cloud feedbacks

Across the historical record with a warming climate, the cloud trends in CMIP5 models have been shown to be in agreement with satellite observations, with robust reductions in cloudiness across the mid-latitude and tropics, as well as an increase in cloud top height at all latitudes (Norris et al., 2016). Our calculated $4xCO_2$ atmospheric albedo feedbacks are consistent with these observations, generally showing a reduction in $\alpha_{atm}$

in the mid-latitudes and tropics (Figure 2). Only one model (GISS) shows an increase in $\alpha_{atm}$ across the $4xCO_2$ simulations.

Most of the $4xCO_2$ GCM-diagnosed $\alpha_{atm}$ feedbacks seem to suggest an increase in $\alpha_{atm}$ in the high-latitudes with warming (particularly over the Southern Ocean), which is likely related to a poleward shift in the storm tracks due to warming (Lu et al., 2007; Norris et al., 2016).

     The $4xCO_2$ GCM-derived $OLW_{cloud}$ feedbacks are also most prominent in the tropics with considerable variability in the location, magnitude and direction of peak feedback (Fig. 4). However, all models show a negative $OLW_{cloud}$ feedback

across the equatorial Pacific and a positive $OLW_{cloud}$ feedback over the Indonesia Archipelago, South America and off the equator. Outside of the tropics, most models show positive $OLW_{cloud}$ feedbacks in the mid-latitudes and slight negative feedbacks in the polar regions. These data are consistent with observations of increased cloud top height (Norris et al., 2016), as regions with enhanced cloudiness (increased $\alpha_{atm}$, Figure 2) also typically show decreased $OLW$ (Figure 4).

     For the LGM, GCM-derived cloud feedbacks are less coherent. Nearly all models show large changes in the tropical

$\alpha_{atm}$ feedback, particularly across the equatorial Pacific and Indonesian Archipelago (Figure 3). Such changes may be suggestive of changes in the position of the Intertropical Convergence Zone (ITCZ) associated changes in deep convective cloud systems that are specific to each model (Braconnot et al., 2007; Arbuszewski et al., 2013). In addition, nearly all GCM-derived feedbacks show a reduction in $\alpha_{atm}$ over the North Atlantic (note that LGM cooling indicates that direction of feedback change is opposite that shown in Figure 3), which may be indicative of a shift in the position of the Gulf Stream seen in some

models (Otto-Bliesner et al., 2006). The prominent feature in the LGM GCM-derived OLW cloud feedback is a large reduction in the tropics (green-blue-purple colors in Figure 5), which is likely related to the reduction in tropical convection due to lower sea surface temperatures (Yin and Battisti, 2001). However, this spatial extent and magnitude of reduction in $OLW_{cloud}$ for the LGM vary appreciably among the GCMs.





### 3.2 Radiative balance in UVic-CFE 4xCO₂ simulation

To compare the global radiative balance of UVic-CFE with that of the GCMs, we calculate the total change in TOA shortwave and longwave fluxes per global mean surface temperature change from the final 10 years of the 150-year 4xCO₂ simulations (relative to the control simulation) and compare the raw GCM results with our cloud feedback-forced UVic-CFE simulations

(Figure 6). The changes in longwave fluxes include the $CO_2$ forcing, which may differ by ~15% between models (Andrews et al., 2012). Because the forcing is included in the longwave fluxes, the flux/temperature ratios shown in Fig. 6 are not a true "feedback," strictly speaking. However, variations in the forcings are presumably relatively small compared to variations in feedbacks. The shortwave flux/temperature ratios in Fig. 6 are true feedbacks and consistent with numbers reported previously (Tomassini et al., 2013).

In general, the spread of TOA shortwave and longwave feedbacks in the 4xCO₂ UVic-CFE simulations matches that of the original GCM results (Figure 6) and is consistent with previous work (Tomassini et al. 2013). For instance, the IPSL model exhibits the largest positive shortwave feedback and largest negative longwave feedback in the GCM results, which is also captured in our UVic-CFE simulations (Figure 2, 4). Conversely, the GISS model is the only simulation to show a negative shortwave feedback and positive longwave feedback, which is consistent with the UVic-CFE results. All other GCM and

UVic-CFE simulations have positive shortwave feedbacks and negative longwave feedbacks that are both smaller in magnitude than the IPSL-based simulations.

    While the relative magnitude of the UVic-CFE feedback results captures that of the original GCM results, the absolute magnitude of the feedbacks is generally slightly reduced in UVic-CFE. We also present the results from a control 4xCO₂ UVic simulation, without the implementation of any cloud feedbacks (grey bar, Figure 6). Here, the TOA shortwave feedback is

~0.40 W m$^{-2}$ °C$^{-1}$ and the TOA longwave feedback is ~-0.03 W m$^{-2}$ °C$^{-1}$, while the average feedbacks from the GCMs are ~0.87 W m$^{-2}$ °C$^{-1}$ and ~-0.55 W m$^{-2}$ °C$^{-1}$, respectively. Therefore, the application of $\alpha_{atm}$ and OLW-cloud feedbacks in UVic-CFE are prominent drivers in the spread of total TOA shortwave and longwave feedbacks. In general, the GCMs show a greater reduction in global surface albedo with increasing temperature compared to the UVic-CFE (not shown). Therefore, the differences in surface albedo processes between the GCMs and UVic-CFE, likely explains some of the reduction in TOA

shortwave feedback magnitude in the UVic-CFE simulations.

### 3.3 Radiative balance in UVic-CFE LGM simulations

For the UVic-CFE LGM simulations, we calculate TOA shortwave and longwave feedbacks at equilibrium conditions, averaged over the last 100 years of the LGM and ctlLGM experiments. Note that in this case the shortwave fluxes include forcing from prescribed ice sheets and therefore are not strictly speaking feedbacks. UVic-CFE generally captures the spread

of the shortwave and longwave feedbacks from the GCMs although it is slightly reduced (Figure 6). The total imbalance seems to be smaller in UVic-CFE compared with most GCMs indicating that UVic-CFE is closer to equilibrium, perhaps because it



was integrated longer. Thus a larger remaining imbalance could contribute to the larger spread in the GCMs compared with UVic-CFE.

The absolute magnitude of the feedbacks is mostly reduced in the UVic-CFE relative to the GCM simulations. Similar to the 4xCO$_2$ results, the IPSL-based simulations present the strongest shortwave and longwave feedbacks. Conversely, the
CNRM-based UVic-CFE simulation shows enhanced shortwave and longwave feedbacks relative to those of the GCM, suggesting that non-cloud processes or differences in the forcings are likely important for this model.

### 3.4 Effect of UVic-CFE on modeled temperature evolution and spatial distribution

As expected, the incorporation of cloud feedbacks into UVic-CFE has a direct impact on modeled surface temperature anomalies in perturbed experiments. For the 4xCO$_2$ experiments, global mean surface air temperature anomalies at the end of
the 150-year simulation range from +3.9 °C (GISS) to +8.8 °C (IPSL), where the control UVic simulation without cloud feedbacks results in a final anomaly of +5.1°C (Figure 7). Only two UVic-CFE simulations (GISS and MRI) result in a year 150 temperature anomaly that is less than the UVic control, confirming that the 4xCO$_2$ net cloud feedbacks are generally positive (see above) and consistent with the analysis of the individual models themselves (Vial et al., 2013; Tomassini et al., 2013).
The spatial variability in GCM cloud feedbacks (Figure 2, 4) is also expressed in the 4xCO$_2$ zonal mean temperature anomalies (Figure 7). All models show the effects of strong polar amplification by the end of the 4xCO$_2$ simulations, but the addition of cloud feedbacks to UVic-CFE appears to enhance this polar amplification in most cases. In addition, the change in temperature due to cloud feedbacks is not uniform for all models. For example, the CCSM-driven simulation presents some of the largest temperature anomalies in the southern high-latitiudes but relatively reduced anomalies at the low-latitudes, resulting
in an overall global anomaly that is similar to the that of the control UVic simulation (Figure 7).

For the LGM simulations, the global mean temperature change at the end of the simulation ranges from -4.1 °C (CCSM) to -8.2 °C (CNRM), whereas the control UVic simulation has a cooling of 5.7°C (Figure 7). Nearly half of the UVic simulations show enhanced global mean cooling (CNRM, IPSL, and MRI) relative to the UVic control (Figure 7), while the other four simulations show reduced cooling (CCSM, GISS, MIROC, and MPI). Again, zonal mean temperature anomalies at
the LGM show that enhanced cloud feedbacks lead to enhanced polar amplification, but spatial differences in the magnitude of feedbacks may impact regional temperature change. For example, the CNRM-based simulation shows the strongest cooling in the southern high latitude, whereas the IPSL-based simulation has the largest cooling in the northern high latitudes (Figure 7).

### 3.5 Using UVic-CFE to estimate climate sensitivity

Intermodel spread in GCM cloud feedbacks has been shown to have a large impact on the modeled sensitivity to perturbation in greenhouse gas radiative forcing (Fasullo and Trenberth, 2012; Andrews et al., 2012; Sherwood et al., 2014). To estimate the effect of the cloud feedbacks in UVic-CFE on global climate, we calculate effective equilibrium climate sensitivity





($\Delta T_{2xC,eff}$) from the 150-year $4xCO_2$ simulations by regressing the global net downward heat flux at the TOA onto the change in temperature. The slope of this regression is the climate response parameter ($\alpha$) and the intercept is the $4xCO_2$ forcing ($F_{4xCO2}$) specific to each model (Gregory et al., 2004). These values can be used to estimate the effective equilibrium climate sensitivity to a doubling of $CO_2$ by dividing the implied global $2xCO_2$ forcing ($F_{2xCO2} = F_{4xCO2}/2$) by $\alpha$ (Gregory et al., 2004). We calculate

$\Delta T_{2xC,eff}$ for both the raw GCM model output as well as the associated UVic-CFE simulations.

With the introduction of cloud feedbacks, UVic-CFE is able to capture much of the intermodel variability in climate sensitivity (Figure 8). The seven GCMs sampled in this analysis show values of $\Delta T_{2xC,eff}$ ranging from 2.15 °C (GISS) to 4.10 °C (IPSL), which agrees well with Andrews et al. (2012) for those models that were used in both studies. In the UVic-CFE simulations, $\Delta T_{2xC,eff}$ values range from 2.34 °C (GISS) to 7.00 °C (IPSL). Again, the IPSL-based UVic-CFE simulation is a

noticeable outlier, while all of the values of $\Delta T_{2xC,eff}$ in UVic-CFE are more comparable to the values from the raw GCM output and the magnitude relative to each of the models is generally the same (Figure 8). However, most of the UVic-CFE simulations show elevated $\Delta T_{2xC,eff}$ relative to their GCM counterpart (Figure 8). The $\Delta T_{2xC,eff}$ in the $4xCO_2$ control UVic simulation (grey bar, figure 8) is 3.63 °C, a value that is higher than most of the GCM results, suggesting that the control UVic climate sensitivity without explicit cloud feedbacks may already be higher than that of most of the sampled GCMs. This

suggests that the control UVic model's clear sky (without explicit clouds) feedbacks are larger than those of most GCMs. Adding the mostly positive cloud feedbacks thus makes the UVic model's climate sensitivities considerably larger than those of the GCMs. Clear-sky feedbacks in the UVic model could be tuned by e.g. varying the coefficients of eq. (6) if a better match with individual GCM's climate sensitivity was desired.

## 4 Discussion and Conclusions

The cloud feedbacks ($\alpha_{atm}$ and $OLW_{cloud}$ feedbacks) derived from the GCMs and employed in UVic-CFE are generally consistent between climate states ($4xCO_2$ vs LGM) for each GCM, with some notable exceptions. For example, the $4xCO_2$ $\alpha_{atm}$ feedbacks (Figure 2) are generally consistent between models in showing a prominent negative feedback across the southern ocean, with CCSM being the only model with a positive $\alpha_{atm}$ feedback. However, for the LGM, the CCSM-derived $\alpha_{atm}$ feedback is negative along with all other models in general (Figure 3). In addition, the $\alpha_{atm}$ feedbacks across the equatorial

Pacific are not always consistent between climate states, with the CNRM-, GISS-, MIROC-, and MPI-based fields showing a pronounced difference in the direction of the $\alpha_{atm}$ feedback (Figure 2, 3). Similarly, the $OLW_{cloud}$ feedbacks across the equatorial Pacific and North Pacific differ in magnitude and direction between the climate states in nearly all models (Figure 4, 5). These differences likely arise due to shifts in the ITCZ and Gulf Stream between climate states (Otto-Bliesner et al., 2006; Braconnot et al., 2007; Arbuszewski et al., 2013), and they suggest that such cloud feedbacks are not universal to all climate states. As

such, the cloud feedbacks derived from the GCMs should only be applied to a consistent climate state experiment when using UVic-CFE.





In general, the application of GCM-derived cloud feedbacks to UVic-CFE captures the changes in TOA radiative balance of the original GCMs, for both the $4xCO_2$ and LGM experiments. Differences in total radiative feedbacks between each GCM and the associated UVic-CFE may exist for several reasons. First, the derivation of the cloud feedbacks are parameterized from the original GCM results and therefore may not be a perfect representation of the full complexity of cloud

radiative forcing in each GCM. This is particularly the case for the shortwave cloud feedback, which is applied using a calculation of the $\alpha_{atm}$ feedback, which uses an assumption of a global mean atmospheric transmissivity (Eq. (3)). The OLW-cloud feedbacks, on the other hand, are a direct calculation of the longwave cloud feedbacks from each GCM.

Second, total TOA feedbacks in UVic-CFE may not perfectly match those of the source GCMs because the resulting feedbacks are still partly controlled by the control radiative balance code of the UVic model. Other components of the Earth

system, apart from clouds, impact the shortwave and longwave radiative balance in UVic, which may feedback on the simulated climate in a different manner than in the GCMs. For instance, the total TOA shortwave feedbacks include the effect of surface albedo change. Therefore, differences in vegetation and sea ice dynamics and their effect on surface albedo in the GCMs relative to UVic may help explain some of the differences in the shortwave feedbacks. Similarly, the longwave feedback in UVic is in part controlled by the SAT-based parameterization of OLW in Eq. 6, which may be different from the clear-sky

feedbacks in the GCMs

Third, the ratios of TOA flux and temperature changes shown in Fig. 6 include forcings (greenhouse gas for both 4xCO2 and LGM and surface albedo for LGM). Therefore, differences in the forcings would also impact the total TOA "feedbacks". The forcings differ between the GCMs but are constant among the UVic-CFE experiments, which may explain some of the loss of spread in UVic-CFE compared with the GCMs.

However, despite the potential for differences in total radiative feedbacks, our results suggest that a simple parameterization of cloud shortwave and longwave feedbacks may be applied to UVic to generally capture dominant inter-model spread in total radiative feedbacks. This result confirms that cloud feedbacks dominate the multi-model uncertainty in GCM radiative balance (Soden and Held, 2006; Dufrense and Bony, 2008; Tomassini et al., 2013; Vial et al., 2013). The addition of GCM-derived cloud feedbacks to the UVic leads to only small increases in computational expense, while capturing

an important component of the Earth's radiative balance that is otherwise lacking in the default UVic model. Indeed, the inclusion of cloud feedbacks leads to a large spread in surface air temperature anomalies for both the $4xCO_2$ and LGM experiments (Figure 7). In addition, spatial variability in the cloud feedbacks (Figure 2-5) leads to some differences in the latitudinal distribution of this temperature change (Figure 7), suggesting that certain regional cloud changes may be important on the global scale. Differences in equator-pole temperature contrast do to cloud feedbacks in UVic-CFE could impact ocean

heat transport in the model.

The application of cloud feedbacks in UVic-CFE provides an important source of inter-model uncertainty that is present in CMIP5/PMIP3. Recent model-data comparisons suggest that the state-of-the-art CMIP5 simulations capture important cloud feedbacks across the observational record (Norris et al., 2016), providing assurance that the feedbacks in UVic-CFE are also within the range of observations. However, as model physics of cloud dynamics and spatial distribution



continue to improve in future GCM simulations, the GCM cloud radiative effects can again be applied in UVic-CFE ensemble analyses to emulate the multi-model uncertainty in cloud feedbacks.

Finally, we confirm that the cloud feedbacks in each of the GCMs plays a prominent role in determining the resulting climate sensitivity of each simulation (Fasullo and Trenberth, 2012; Andrews et al., 2012; Sherwood et al., 2014). By incorporating cloud feedbacks into UVic-CFE, we generally capture the relative spread $\Delta T_{2xC,eff}$ of the GCMs (Figure 8). The absolute magnitude of $\Delta T_{2xC,eff}$ is typically larger in our UVic-CFE simulations relative to each of the GCMs. Since net cloud feedbacks are generally positive in CMIP5 (Vial et al., 2013; Tomassini et al., 2013), the addition of these radiative feedbacks may require a revision of the overall radiative balance in UVic-CFE, particularly through an enhanced OLW parameterization by slight adjustment to the constants in Eq. (6). This method of has been applied to UVic to effectively adjust $\Delta T_{2xC,eff}$ (Schmittner et al., 2011). The UVic-CFE is currently being applied to a study of climate sensitivity using paleoclimate reconstructions (Ullman et al., in prep.).

**Code and Data Availability**

UVic-CFE model code, associated cloud feedback input files, and other relevant data files are available as a Supplement to this manuscript. See the README file in the Supplement for description of contents.

**Acknowledgements**

This work was supported by a grant from the National Science Foundation's Paleoclimate Perspectives on Climate Change (P2C2) program (award number 1204243).

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





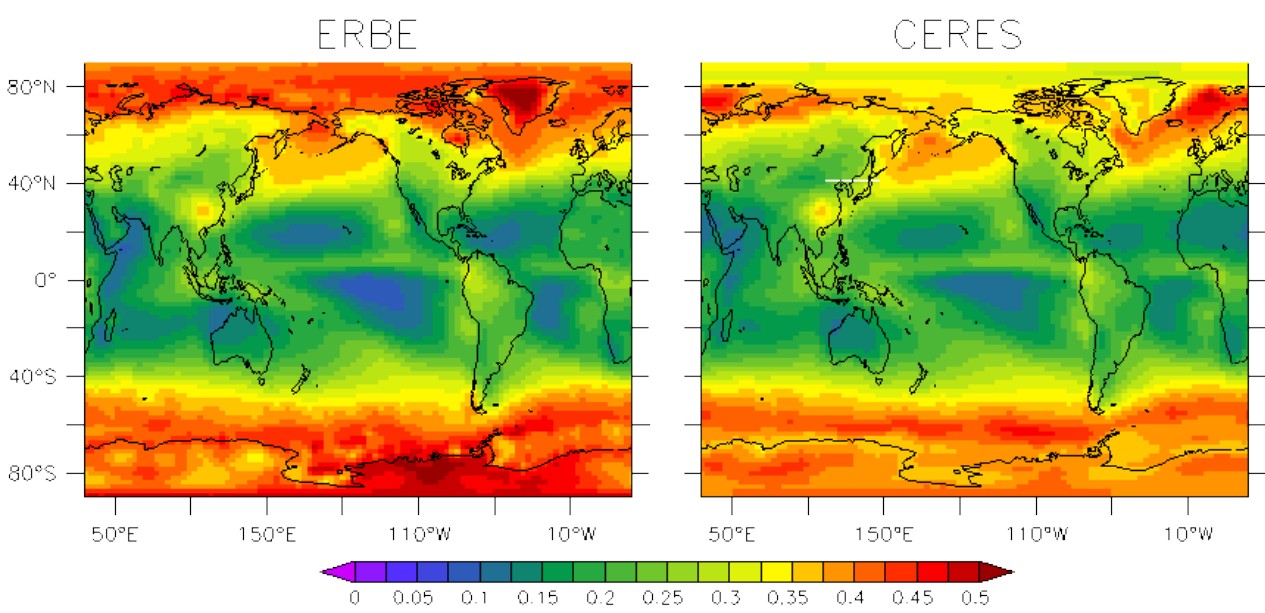

**Figure 1: Comparison of annual-averaged atmospheric albedo ($\alpha_{atm}$) as calculated using Eq. (3) and the climatology of ERBE (left) and CERES (right) data.**





5  **Figure 2: Maps of annual-mean atmospheric albedo feedback term ($\alpha_{atm}FB$), as calculated using Eq. (11) and the 4xCO₂ results of the 7 CMIP5 models discussed in the text. Units are albedo fraction change per °C.**



**Figure 3: Maps of annual-mean atmospheric albedo feedback term ($\alpha_{atm}FB$), as calculated using Eq. (11) and the LGM results of the 7 PMIP3 models discussed in the text. Units are albedo fraction change per °C. Note that because the LGM represents a period of global cooling (Braconnot et al., 2012), the direction of change in $\alpha_{atm}$ is opposite that shown in these figures.**







**Figure 4: Maps of annual-mean outgoing longwave feedback term (*OLW_cloud FB*), as calculated using Eq. (14) and the 4xCO₂ results of the 7 CMIP5 models discussed in the text. Units are W m⁻² °C⁻¹.**



**Figure 5: Maps of annual-mean outgoing longwave feedback term ($OLW_{cloud}FB$), as calculated using Eq. (14) and the LGM results of the 7 PMIP3 models discussed in the text. Units are W m$^{-2}$ °C$^{-1}$. Note that because the LGM represents a period of global cooling (Braconnot et al., 2012), the direction of change in $OLW_{cloud}$ is opposite that shown in these figures.**





**Figure 6: Comparison of 4xCO₂ (top) and LGM (bottom) top-of-the-atmosphere feedbacks calculated from raw CMIP5/PMIP3 output from each of the 7 GCMs (CMIP5/PMIP3) and from UVic simulations using GCMs-derived cloud feedbacks (UVic). Shortwave feedbacks are shown on the left, longwave feedbacks on the right. Positive values designate an increased forcing *TO* the climate system with increased temperature (i.e. positive feedback). Feedbacks from the UVic control simulation without cloud feedbacks is shown in grey.**





5    **Figure 7: Global mean surface air temperature anomalies for the 4xCO₂ (upper left) and LGM (upper right) UVic-CFE simulations. Zonal mean surface air temperature anomalies from the UVic-CFE simulations, averaged over the last 10 years of the 4xCO₂ simulations (lower left) and the last 100 years of the LGM simulation (lower right).**





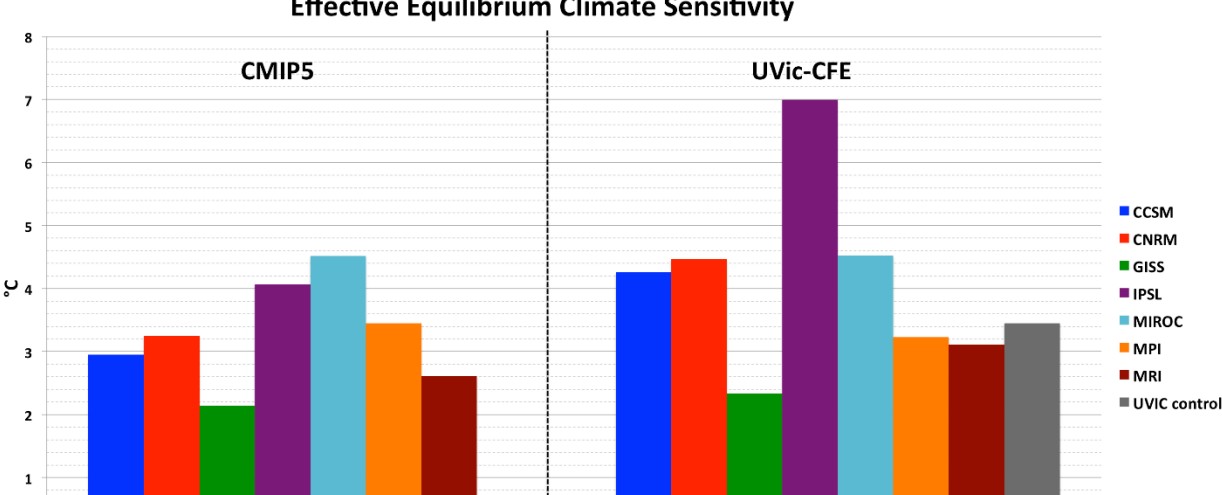

**Figure 8: Comparison of effective equilibrium climate sensitivity ($\Delta T_{2xC,eff}$) calculated from raw CMIP5 output from each of the 7**
5     **GCMs (CMIP5) and from UVic simulations using GCMs-derived cloud feedbacks (UVic). $\Delta T_{2xC,eff}$ from the UVic control simulation without cloud feedbacks is shown in grey.**