# Peer review of "A Cloud Feedback Emulator (CFE, version 1.0) for an intermediate complexity model"

_Geoscientific Model Development, 2016_

## Short Comment (SC1) · 26 Sep 2016

Dear authors,

In my role as Executive editor of GMD, I would like to bring to your attention our Editorial version 1.1:

http://www.geosci-model-dev.net/8/3487/2015/gmd-8-3487-2015.html

This highlights some requirements of papers published in GMD, which is also available on the GMD website in the 'Manuscript Types' section:

http://www.geoscientific-model-development.net/submission/manuscript_types.html

In particular, please note that for your paper, the following requirement has not been met in the Discussions paper:

- "The main paper must give the model name and version number (or other unique identifier) in the title."

Please add a version number for your model in the title upon your revised submission to GMD.

Yours,

Astrid Kerkweg

---

## Referee Comment (RC1) · Anonymous Referee #1 · 11 Oct 2016

General comments:

The paper "The University of Victoria Cloud Feedback Emulator (UVic-CFE): cloud radiative feedbacks in an intermediate complexity model" describes and evaluates a new method for applying GCM-derived cloud feedbacks to intermediate complexity models. The new method is able to capture the spread in TOA radiative feedbacks between the original GCMs, implying that the tool is generally efficient. Given that cloud feedback plays an essential role in determine the magnitude of global warming, this method is expected to be useful in improving intermediate complexity models that are in lack of cloud feedbacks. Therefore, I suggest the paper be published after addressing my following comments.

1. The cloud masking effect has not been removed from cloud radiative effect when cloud feedback is calculated. This may result in a systematic bias to the TOA radiations.

Consider an assumptive situation that there is no change in cloud properties under global warming, then no cloud terms need to be added up to Eq. (6), and the OLW_cloud(t) term in Eq. (15) should be zero. However, the cloud longwave radiative effect in GCM would still change due to changes in water vapor and temperature (cloud masking effect), leading to a non-zero value in Eq. (15). Therefore, an additional term is needed to compensate the cloud masking effect (this may be done with radiative kernels, Soden et al. 2008, doi: 10.1175/2007JCLI2110.1).

2. The calculation of "TOA feedbacks" is inaccurate, so I suggest the authors to either calculate the TOA feedbacks with the standard method, or to replace "TOA feedbacks" with another phrase. In this paper, climate forcing is included in the "TOA feedbacks" (Page 9, line 6), so the TOA feedbacks in Fig. 6 ($\sim$0W/m2/K) is much larger than that calculated by Andrews et al. 2012 (-1.08 W/m2/K).

3. The cloud rapid adjustment has not been removed from the cloud feedback (Zelinka et al. 2013, doi: 10.1175/JCLI-D-12-00555.1). This may be partially responsible to the loss of spread in UVic-CFE simulations (Fig. 6).

I expect that the UVic-CFE would be more accurate after the above comments are addressed.

Specific comments:

Page 1, Line 29. "The relative magnitude and net effect of these feedbacks depends on cloud altitude... High clouds, on the other hand, radiate at much colder temperatures than the surface, which can make the longwave effect dominate and lead to net warming"

The authors may also discuss the effect of cloud optical depth here. The net cloud radiative effect of high clouds could be either positive (high thin clouds) or negative

(high thick clouds) depending on the cloud optical depth.

Page 3, Line 30. It is worth discussing whether the cloud radiative effect is included when the empirical parameters in Eq. (6) were calculated.

Page 7, Line 17. Please provide more details for the 4xCO2 and LGM experiment design. Figure 4 and 5. There are some white pixels surrounded by blue/green pixels (for example, Central tropical Pacific Ocean in Fig. 4a,d). Are these white pixels induced by missing values? Is it possible to eliminate them?

Technical comments

Page 3, Line 25. How is the variable f calculated?

Figure 6, 8. Figure legend: "UVIC control" -> "UVic control", to be consistent with the figure description.

---

## Referee Comment (RC2) · Anonymous Referee #2 · 14 Oct 2016

From the reference to the paper by Weaver and co-authors, UVic needs a choice between options, like humidity transport or diffusive. It also looks as if the atm wind can be prescribedd or sensitive to SAT and density. Evidently, such options impact the feedbacks in UVic. I suggest a table to summarize these options, and the feedback concerned..

In the same views, the way cloud feedback, as approximated by the atm-albedo could be explicitly described for clarity. The description of some feedback-loops would be of great help. For instance, atm-albedo -> SAT-> OLW etc How is the ocean dynamics impacted ? What changes are observed concerning the thermohaline circulation, the thermocline etc What about the sea-ice extent?

Some comments are already included in the RESULTS sections that could be related more closely to the UVic extended results others that the averaged global results directly as support to the comparison with the seven GCMs

---

## Editor Comment (EC1) · K. Gierens (Editor) · 7 Nov 2016

I have a few comments/questions concerning several equations in the manuscript. Please consider them for your revised version.

1) Eq. 3: a better explanation of $f$ is needed. How does the choice of $f$ guarantee that $\alpha_{atm}$ remains bounded by 0 and 1? Also it should be stated that $S$ in equation 2 is identical to $SW_{in,TOA}$ in eq. 4.

2) Eqs. 11 and 12: The argument that albedo values are not additive leads you to formally consider the ratio $\alpha_{atm,perturbed}/\alpha_{atm,CERES}$ in eq. 11, however it is necessary to subtract one from this ratio. Mathematically, we then have the difference of the

albedo values back, since

$$(\alpha_{atm,perturbed}/\alpha_{atm,CERES}) - 1 = (\alpha_{atm,perturbed} - \alpha_{atm,CERES})/\alpha_{atm,CERES}.$$

In eq. 12 this expression is then multiplied by $\alpha_{atm,CERES}$, and the simple difference of the albedo values returns back. So this argumentation seems to add unnecessary complexity.

3) Eqs. 12 and 15: I wonder whether these equations are used at every timestep. If so, how do you distinguish climatological temperature variations from diurnal and seasonal temperature variations? Should a feedback not work only on the long climatological time scales? Furthermore, are these equations applied to each grid point independently or are they averaged over, e.g., latitude zones?

4) Page 11, line 4: Why do you write $F_{2\times CO_2} = F_{4\times CO_2}/2$ when there is a logarithmic relation between radiative fluxes and the $CO_2$ concentration? Is this close to linear because the absolute change is very small?

————————————

---

## Author Comment (AC1) · 17 Jan 2017

Thank you for bringing this requirement to our attention. Since our model is a modification to UVic v 2.9, we have chosen to remove this distinction from the title. Therefore, we have revised our title to read:

A Cloud Feedback Emulator (CFE version 1.0) for an intermediate complexity model

---

## Author Comment (AC2) · 18 Jan 2017

**Response to Anonymous Referee #1**

We thank Anonymous Referee #1 for his/her thoughtful and insightful comments on our manuscript. We have responded to the comments below (in red).

General comments:
The paper "The University of Victoria Cloud Feedback Emulator (UVic-CFE): cloud radiative feedbacks in an intermediate complexity model" describes and evaluates a new method for applying GCM-derived cloud feedbacks to intermediate complexity models. The new method is able to capture the spread in TOA radiative feedbacks between the original GCMs, implying that the tool is generally efficient. Given that cloud feedback plays an essential role in determine the magnitude of global warming, this method is expected to be useful in improving intermediate complexity models that are in lack of cloud feedbacks. Therefore, I suggest the paper be published after addressing my following comments.

1. The cloud masking effect has not been removed from cloud radiative effect when cloud feedback is calculated. This may result in a systematic bias to the TOA radiations. Consider an assumptive situation that there is no change in cloud properties under global warming, then no cloud terms need to be added up to Eq. (6), and the OLW_cloud(t) term in Eq. (15) should be zero. However, the cloud longwave radiative effect in GCM would still change due to changes in water vapor and temperature (cloud masking effect), leading to a non-zero value in Eq. (15). Therefore, an additional term is needed to compensate the cloud masking effect (this may be done with radiative kernels, Soden et al. 2008, doi: 10.1175/2007JCLI2110.1).

Because the cloud masking effect is evident in the $4xCO_2$ GCM simulations, we choose to include it in our OLW cloud feedback. The OLW parameterization of UVic (Eq. 6) lacks the full cloud dynamics of the GCMs, so by assessing our OLW cloud feedback with the full cloud radiative effect (CRE) anomalies, we aim to incorporate the full OLW cloud changes of the GCMs, including cloud-masking effects. That said, the Referee is correct in pointing out the differences in the CRE and kernel methods, so we make note of potential issues as suggested by the reviewer (section 2.3.2).

2. The calculation of "TOA feedbacks" is inaccurate, so I suggest the authors to either calculate the TOA feedbacks with the standard method, or to replace "TOA feedbacks" with another phrase. In this paper, climate forcing is included in the "TOA feedbacks" (Page 9, line 6), so the TOA feedbacks in Fig. 6 (_0W/m2/K) is much larger than that calculated by Andrews et al. 2012 (-1.08 W/m2/K).

Since the exact greenhouse gas forcing is unknown for each of the GCMs, we favor our current method of calculating TOA "feedbacks". Therefore, we have replaced the phrase "TOA feedbacks" with "TOA radiative-temperature response" so as not to confuse readers with the standard method.

3. The cloud rapid adjustment has not been removed from the cloud feedback (Zelinka et al. 2013, doi:10.1175/JCLI-D-12-00555.1). This may be partially responsible to the loss of spread in UVic-CFE simulations (Fig. 6).I expect that the UVic-CFE would be more accurate after the above comments are addressed.

Yes, by calculating our cloud radiative effect using CRE anomalies instead of the 'Gregory' slope method, we are neglecting influence of cloud rapid adjustment. However, we are unable to include them as part of the forcing (as suggested by Zelinka et al., 2013), so we must include them in the cloud feedback. We make note of this issue in the revised text (discussion) and its possible influence on the spread in UVic-CFE.

Specific comments:
Page 1, Line 29. "The relative magnitude and net effect of these feedbacks depends on cloud altitude. . . High clouds, on the other hand, radiate at much colder temperatures than the surface, which can make the longwave effect dominate and lead to net warming"

The authors may also discuss the effect of cloud optical depth here. The net cloud radiative effect of high clouds could be either positive (high thin clouds) or negative (high thick clouds) depending on the cloud optical depth.
Yes, the referee is correct in noting the importance of optical depth. We now include a statement on cloud optical depth and its relationship with cloud radiative effect.

Page 3, Line 30. It is worth discussing whether the cloud radiative effect is included when the empirical parameters in Eq. (6) were calculated.

The original parameterization of Thompson and Warren (1982) is for clear-sky outgoing LW fluxes. Therefore, there is no cloud feedback within this parameterization. We now make note of this in our description of Eq. (6).

Page 7, Line 17. Please provide more details for the 4xCO2 and LGM experiment design.

Our experimental design is described in the second paragraph of section 2.4. Is there a particular aspect of the experimental design that you would like to see further described?

Figure 4 and 5. There are some white pixels surrounded by blue/green pixels (for example, Central tropical Pacific Ocean in Fig. 4a,d). Are these white pixels induced by missing values? Is it possible to eliminate them?

These white pixels appear to be due to an improper image rendering of the original postscript image. We have corrected these images by first saving the file to a jpeg.

Technical comments
Page 3, Line 25. How is the variable f calculated?

The variable f is as an adjustment factor that is necessary to correct for a radiative imbalance that arises in our estimate of atmospheric albedo from the CERES observational data. Therefore, this adjustment factor directly affects global mean surface air temperature. We ran a series of control simulations with different f-values to tune our control simulation to observational estimates of ~13.9 C (f=0.95). We have rephrased our description in section 2.2 to address this confusion.

Figure 6, 8. Figure legend: "UVIC control" -> "UVic control", to be consistent with the figure description.

Thank you for this catch. We have fixed this in our updated manuscript.

---

## Author Comment (AC3) · 18 Jan 2017

**Response to Anonymous Referee #2**

We thank Anonymous Referee #2 for his/her thoughtful and insightful comments on our manuscript. We have responded to the comments below (in red).

From the reference to the paper by Weaver and co-authors, UVic needs a choice between options, like humidity transport or diffusive. It also looks as if the atm wind can be prescribed or sensitive to SAT and density. Evidently, such options impact the feedbacks in UVic. I suggest a table to summarize these options, and the feedback concerned.

The referee is correct in noting that UVic contains options regarding atmospheric transport/diffusion and atm winds. In the first paragraph of section 2.1 (4th sentence), we have stated that we are using UVic version 2.9, which includes the atmospheric heat diffusion feedback (diffusion as a function of global mean surface air temperature). This is a feedback that the latest version of UVic includes and we make note that it has been shown to improve the latitudinal temperature gradient for the Last Glacial Maximum (when compared with proxy data; Fyke and Eby, 2012).

To isolate the effect of cloud feedbacks in our emulator, we choose to prescribe atm wind stress (no SAT feedback). However, large differences in the surface boundary conditions at the LGM (ice sheets) have been shown to greatly impact wind stress anomalies in LGM simulations (Muglia and Schmittner, 2016). The optional wind-SAT feedback would not capture these changes; therefore, we apply wind stress anomalies as diagnosed from the LGM GCM results (see end of section 2.4).

To be consistent in our model design, we also prescribe modern wind stress for our 4xCO2 simulations. Wind stress anomalies across the CMIP5 4xCO2 experiments are small; therefore, we use the prescribed wind stress fields of the control UVic 2.9 model. Upon reviewing our manuscript, we discussed the use of LGM wind stress anomalies at the end of section 2.4, but did not note our wind stress boundary condition for the 4xCO2 simulations. We have added an additional sentence at the end of section 2.4 that further discusses the prescribed wind stress for the 4xCO2 simulations.

In the same views, the way cloud feedback, as approximated by the atm-albedo could be explicitly described for clarity. The description of some feedback-loops would be of great help. For instance, atm-albedo -> SAT-> OLW etc How is the ocean dynamics impacted? What changes are observed concerning the thermohaline circulation, the thermocline etc What about the sea-ice extent?

We provide a description of the nature of cloud feedback loops (through their impact on SW and LW radiative balance) in our introduction.

Regarding ocean dynamics and sea-ice, we have chosen to concentrate this manuscript on how our linear parameterization of cloud feedbacks helps capture the change in radiative balance that would otherwise be missing in UVic (or similar EMIC). We have focused our discussion of radiative feedbacks on surface air temperature evolution and climate sensitivity, by association with the $4xCO_2$ experiments of the CMIP5 coordinated framework. Therefore, we have not included any analysis on the impacts of these radiative changes on ocean dynamics in this study.

Some comments are already included in the RESULTS sections that could be related more closely to the UVic extended results others that the averaged global results directly as support to the comparison with the seven GCMs.

See responses to comments from other reviewer/editors.

---

## Author Comment (AC4) · 18 Jan 2017

**Response to Editor Comment 1: K. Gierens**
Thank you to the Editor for the comments. Our response is written below (in red).

I have a few comments/questions concerning several equations in the manuscript. Please consider them for your revised version.

1) Eq. 3: a better explanation of f is needed. How does the choice of f guarantee that $\alpha_{atm}$ remains bounded by 0 and 1? Also it should be stated that S in equation 2 is identical to $SW_{in,TOA}$ in eq. 4.

For explanation of f, see response to Referee #1. We have rephrased our description in section 2.2 to address this confusion.

The variable f does not guarantee that $\alpha_{atm}$ is bounded by 0 and 1. However, the editor is correct in noting that such a limit should be in place. We set this limit in our calculation of $\alpha_{plt}$ and $\alpha_{sfc}$. Under low-light conditions (winter, high-latitudes), the denominators of equation 4 and 5 ($SW_{in,TOA}$) can become small relative to the numerator, resulting in a value of $\alpha_{plt} > 1$ or $\alpha_{sfc} > 1$ across a latitudinal band. Therefore, we limit $\alpha_{plt}$ and $\alpha_{sfc}$ to be between 0 and 1. If $\alpha_{plt}$ and $\alpha_{sfc}$ are greater than 1, we assign them with a value from the next closest month in time where $\alpha_{plt}$ and $\alpha_{sfc}$ are appropriately defined. However, we note that these are months with low incoming light ($SW_{in,TOA}$), so the effect of $\alpha_{plt}$ and $\alpha_{sfc}$ on local radiative balance is negligible. In short, we do limit $\alpha_{atm}$ through a limit on $\alpha_{plt}$ and $\alpha_{sfc}$. With these limits, $\alpha_{plt}$ is bounded by 0 and 1. We have added a sentence explaining this in our description of new $\alpha_{atm}$ with CERES data (section 2.2).

We have adjusted S in eq. 2 so that it now uses the consistent variable "$SW_{in,TOA}$" as in eq. 4.

2) Eqs. 11 and 12: The argument that albedo values are not additive leads you to formally consider the ratio $\alpha_{atm,perturbed}/\alpha_{atm,CERES}$ in eq. 11, however it is necessary to subtract one from this ratio. Mathematically, we then have the difference of the albedo values back, since
$(\alpha_{atm,perturbed}/\alpha_{atm,CERES}) - 1 = (\alpha_{atm,perturbed} - \alpha_{atm,CERES})/\alpha_{atm,CERES}$.

In eq. 12 this expression is then multiplied by $\alpha_{atm,CERES}$, and the simple difference of the albedo values returns back. So this argumentation seems to add unnecessary complexity.

Yes, the editor is correct in noting that the 1 is mathematically unnecessary. However, in defining the atm albedo feedback this way (centered around zero), it is easier to demostrate when the feedback is positive and when it is negative. By association, we feel the plots of the atm albedo feedback are more illustrative when centered around zero.

3) Eqs. 12 and 15: I wonder whether these equations are used at every timestep. If so, how do you distinguish climatological temperature variations from diurnal and seasonal temperature variations? Should a feedback not work only on the long climatological time scales? Furthermore, are these equations applied to each grid point independently or are they averaged over, e.g., latitude zones?

As stated in section 2.3, these equations are used at every timestep. We assess these cloud radiative feedbacks ($\alpha_{atm}FB$ and $OLW_{cloud}FB$) over a 12-month climatology to incorporate any seasonality in the feedbacks (e.g. monsoon impacts, etc). In addition, $\alpha_{atm}FB$ and $OLW_{cloud}FB$ are applied at each grid cell to incorporate the spatial patterns in the cloud feedbacks that are unique to each source GCM. We have attempted to clarify this in our revised manuscript.

4) Page 11, line 4: Why do you write $F_{2\times CO_2} = F_{4\times CO_2}/2$ when there is a logarithmic relation between radiative fluxes and the $CO_2$ concentration? Is this close to linear because the absolute change is very small?

Yes, there is a logarithmic relationship between CO2 radiative forcing and concentration. In UVic, it looks like this:

$$F_{CO2} = CO2FOR * ln\frac{[CO2\,(ppm)]}{280\,ppn},$$

where CO2FOR is the CO2 radiative forcing term (5.35 W m$^{-2}$), equivalent to 3.71 W m$^{-2}$ for a doubling of CO2.

The forcing of 4xCO2 (1120 ppm) is mathematically equivalent to 2x the forcing of a doubling of CO2 (560 ppm). Therefore, $F_{2xCO2} = F_{4xCO2}/2$